# FairContrast: Enhancing Fairness through Contrastive learning and Customized Augmenting Methods on Tabular Data

## Abstract

As AI systems become more embedded in everyday life, the development of fair and unbiased models becomes more critical. Considering the social impact of AI systems is not merely a technical challenge but a moral imperative. As evidenced in numerous research studies, learning fair and robust representations has proven to be a powerful approach to effectively debiasing algorithms and improving fairness while maintaining essential information for prediction tasks. Representation learning frameworks, particularly those that utilize self-supervised and contrastive learning, have demonstrated superior robustness and generalizability across various domains. Despite the growing interest in applying these approaches to tabular data, the issue of fairness in these learned representations remains underexplored. In this study, we introduce a contrastive learning framework specifically designed to address bias and learn fair representations in tabular datasets. By strategically selecting positive pair samples and employing supervised and self-supervised contrastive learning, we significantly reduce bias compared to existing state-of-the-art contrastive learning models for tabular data. Our results demonstrate the efficacy of our approach in mitigating bias with minimum trade-off in accuracy and leveraging the learned fair representations in various downstream tasks.

## 1 Introduction

The real-world application of deep learning approaches is expanding rapidly. These approaches are susceptible to stereotypes or societal biases inherited in the data, resulting in models biased against individuals with specific sensitive attributes and treating them unfairly. Well-known examples include a recidivism risk prediction model that predicts reoffending rates for individuals of certain races at twice the rate compared to others [1, 9, 33] , or recruitment models showing bias towards male candidates over equally qualified female candidates[29, 11, 10]. Consequently, there is an increasing amount of research focused on algorithmic fairness, with the primary objective being to guarantee that sensitive attributes have no impact on algorithmic outcomes.

Learning fair and robust representations has shown its potential in effectively debiasing and improving fairness while keeping the essential information for the prediction task [44]. In representation learning frameworks, self-supervised learning and particularly contrastive learning have shown superior robustness and generalizability across various domains, including natural language processing (NLP) and computer vision, even in scenarios with fully labeled or few labeled data [4]. Contrastive learning is more challenging when applied to tabular datasets compared to other data types such as images, text, or speech. This is because these datasets typically contain spatial, semantic, and vocal relationships, which provide structured information not present in tabular data. Although contrastive learning for tabular data is receiving increased attention, the fairness in these learned representations has not been

thoroughly explored. Our particular interest lies in investigating whether contrastive learning methods can be utilized to mitigate bias and improve the fairness of representations. We argue that other contrastive learning models for tabular data, such as VIME [45] and SCARF [3], which do not address fairness issues, exhibit bias in their predictions in downstream tasks, leading to discrimination.

In this study, we propose a contrastive learning framework for tabular dataset, to mitigate bias and learn fair representations. For positive pairs, each privileged sample with favorable outcome is paired with one randomly selected unprivileged sample with favorable outcome. Other samples are paired with one randomly selected sample with the same class and the same sensitive attribute. In our supervised and unsupervised framework, supervised contrastive learning loss [27] and InfoNCE loss [17, 32] combined with binary cross-entropy loss are used, respectively. Our proposed method is evaluated on various prevalent datasets in the fairness domain. The results demonstrate a significant reduction in bias compared to existing state-of-the-art contrastive learning frameworks for tabular datasets. Moreover, our framework results in learning fair representations that can be utilized in any downstream tasks.

## 2 Related Work

**Self-Supervised Learning for tabular data.** As access to unlabeled data expands, self-supervised learning (SSL) has received attention across various domains, including natural language processing (NLP), computer vision, and speech recognition. SSL approaches are representation learning frameworks that use unlabeled data to learn robust and meaningful representations. SSL also demonstrates its ability in robustness and generalizability even in scenarios with fully labeled or few labeled data [4]. SSL methods highly depend on the correlations within the features of the data. Recently, there has been an increasing focus within the representation learning field on employing SSL for tabular data. Unlike other data types such as images, text, or speech, which possess distinct structures such as spatial, semantic, and vocal relationships respectively, tabular data are more challenging [42]. This is primarily due to the absence of explicit relationships for learning representation, which also varies across different tabular datasets[42].

Several studies [2, 35, 3, 45, 7, 18, 23, 40, 43] have been proposed that deploy SSL methods in tabular datasets. These approaches can be categorized to two groups : 1. employing the pretext tasks and 2. contrastive learning. Deploying the pretext task is the most widely-used category. [45] proposes Value Imputation and Mask Estimation(VIME), a self and semi supervised framework for tabular data. In the self supervised framework, they pre-trained an encoder on unlabeled corrupted/masked data and extracted the features. Then these representations are passed through "mask estimation" and "feature estimation" heads for recovering the binary mask used for corruption, and the value of the feature that has been masked. This pre-trained encoder is utilized in the semi-supervised learning framework as well. Their choice of the corruption strategy is to choose a mask randomly sampled from a Bernoulli distribution. For the filling strategy they utilize CutMix [46], which replaces all masked values of each sample with values from another randomly selected sample. VIME achieved state-of-the-art on clinical and genomics datasets. On the other hand, the main concept of contrastive learning is to pull similar instances (positive pairs) together and push dissimilar instances (negative pairs) away. This is achieved by maximizing agreement (or minimizing the distance) within the embedding space of positive pairs while maximizing the distances with negative pairs. [8, 36, 32, 14]. These methods have been very successful in computer vision. In data types such as images, the positive pairs can be generated through a data augmentation module (i.e., through transformation of the image such as cropping and resizing, rotation, color dissertation, etc.), while negative pairs correspond to other images in the batch [8]. More recently the scope of contrastive learning has been extended to weakly supervised [38], semi-supervised, and supervised setup. These studies have introduced an extra conditional variable such as details about the downstream task [37] or labels from downstream tasks [27, 26], to improve the quality of the representations. In the supervised contrastive learning setup, the positive pairs belong to the same class while negative pairs belong to different classes [27]. In the NLP domain, it has been demonstrated that the model's robustness to noise and data sparsity can be improved when supervised contrastive learning is combined with a cross entropy loss [15, 34]. SCARF proposed by [3] is another state-of-the-art approach that deploys contrastive learning on tabular data. In the SSL setting Scarf method masks 60% of the features for each data point and then uses Random Feature Corruption to replace these masked values. Finally they fine-tune their encoder weights with a classification head on top through some labeled data. The authors of this

study claim the contrastive learning setting of SCARF is superior compared to pre-trained models such as VIME. They compared their methods with other random feature corruption methods, such as CutMix [46] and MixUp [48] and argue that their proposed random feature corruption method is more effective. As mentioned earlier, CutMix [46] replaces values from one randomly chosen sample for all of the masked values in each sample, whereas SCARF method randomly selects a sample for each masked value. In addition, the corruption method of MixUp [48] is to replace it with a linear combination of the sample and another randomly chosen sample. It is demonstrated that MixUp is more effective when corruption is done on the embedding space rather than input space [35].

**Fair Contrastive learning** Contrastive learning has been extensively used to address fairness concerns in the field of computer vision [47, 22, 5, 39]. In contrast, while the application of self-supervised contrastive learning to tabular data is gaining attention, its use for fairness-specific objectives remains comparatively underexplored. [30] introduces a conditional contrastive learning (CCL) approach primarily designed for vision tasks, which they also extend to tabular data. Their method selects positive and negative pairs conditioned on the sensitive attribute to minimize sensitive leakage while improving class separability. However, their augmentation strategy, adding isotropic Gaussian noise to standardized tabular features, originates from vision-based contrastive frameworks and is less well-suited for tabular data. Many tabular features are discrete, semantically structured, or non-continuous, making Gaussian perturbations potentially unrealistic or semantically meaningless. In the domains of NLP and computer vision, [34] proposed a contrastive learning-based method for bias mitigation that encourages representations of samples with the same class label to be close, while pushing apart representations that share the same protected attribute. Although developed for unstructured data, the underlying principle of this approach, decoupling class and sensitive attribute representations, is also applicable to tabular data. [6] employs self-supervised setting on tabular data in an encoder- decoder framework and discusses fairness; however, they do not utilize contrastive learning. DualFair [19] presents a self-supervised representation learning framework that jointly addresses both group fairness and counterfactual fairness. It achieves this by generating counterfactual samples using a cyclic variational autoencoder (C-VAE), applying fairness-aware contrastive loss to align embeddings across sensitive groups and counterfactuals, and using self-knowledge distillation to maintain representation quality.

## 3   Method

Figure 1 illustrates the general framework of our proposed model. A customized technique is used to selectively pair positive samples from the original input based on specific criteria. These positive pairs are integrated into the training process using a contrastive loss, alongside classification tasks, in an end-to-end manner.

The selection of negative pairs in our contrastive learning framework depends on whether the setting is supervised or self-supervised. In the supervised version, negative pairs within a batch are formed from samples of different classes. In the self-supervised version, all other samples in the batch are treated as negatives.

For positive pairs, instances from the same class are further conditioned on the sensitive attribute. That is, we sample pair instances within the same subgroup (defined by outcome * sensitive attribute), such as Female with low income paired with another Female with low income. The only exception is for the privileged group with a favorable outcome. This design helps preserve subgroup-specific characteristics in the learned representations, ensuring the model remains accurate within subgroups while maintaining clear separation between them.

For the privileged group with favorable outcomes, positive pairs are drawn from the unprivileged group with the same favorable outcome, as the privileged group poses challenges to the classifier's ability to ensure fair predictions. For example, we pair a Male high income instance with a Female high income instance. Keeping them in the same class (high income) ensures that the model learns to minimize representational distance between them. This approach encourages the model not to rely on the sensitive attribute in favorable outcome predictions, aligning with the fairness criterion of equality of opportunity by promoting intra-group similarity across sensitive attributes.

The general idea behind contrastive learning is to train a model to bring similar samples closer together in a learned representation space while pushing dissimilar samples apart. Our strategy embeds fairness into the model's learning process without altering the core contrastive learning

mechanism. By encouraging instances from favorable outcomes with different sensitive attributes to be closer in representation space, we naturally achieve fairness goals without relying on additional fairness-specific constraint-based loss functions. Incorporating our custom sampling strategy and optimizing with contrastive loss encourages the embeddings of selected groups to converge, reducing bias and mitigating discrimination while preserving model utility.

To implement this approach, our architecture includes an encoder $z = Enc(x)$, which maps the input to a representation. These representations are then used to calculate both the contrastive loss and the classification loss. Within this framework, we investigate a range of contrastive loss functions:

- **Self-supervised contrastive loss:** Self-supervised contrastive learning does not require explicit labels for training. Instead, it leverages positive and negative pairs of the data sample. In this context, given a mini-batch with a set of N randomly selected samples, let $i \in \{1...N\}$ be the index of an arbitrary sample, called the *anchor* and let j be the index of random augmentations (a.k.a., "views"), also called the *positive*, the corresponding mini-batch consists of $2N$ pairs where the other $2(N-1)$ indices $\{1...N\} \setminus \{i\}$ are called the *negatives*. Here $z_i = Enc(x_i)$, $\tilde{z}_j = Enc(\tilde{x}_j)$ denotes the embeddings generated from the encoder and the self-supervised contrastive loss is calculated as [8, 36, 21]:

$$L^{self} = -\frac{1}{N} \sum_{i=1}^{N} \log \frac{\exp(\text{sim}(z_i, z_j)/\tau)}{\sum_{k \neq i} \exp(\text{sim}(z_i, z_k)/\tau)} \quad (1)$$

where **sim** function is the Cosine Similarity (Eq. 2).

$$\text{sim}(z_i, z_j) = \frac{z_i^T \tilde{z}_j}{\|z_i\|_2 \cdot \|\tilde{z}_j\|_2} \quad (2)$$

$\tau \in R^+$ is a scalar temperature parameter controlling softness.

- **Supervised contrastive loss:** In the realm of supervised learning, the contrastive loss outlined in Eq. 1 encounters limitations when multiple samples are known to belong to the same class [27]. Eq. 3 presents the most direct approaches for extending Eq. 1 to include supervision [27]:

$$L^{sup} = -\frac{1}{N} \sum_{k=1}^{N} \frac{1}{|P(i)|} \sum_{p \in P(i)} \log \frac{\exp(z_i \cdot z_p/\tau)}{\sum_{q \in Q(i)} \exp(z_i \cdot z_k/\tau)} \quad (3)$$

where the symbol $\cdot$ denotes the inner (dot) product and $Q(i) \equiv \{1...N\} \setminus \{i\}$, $P(i) \equiv \{p \in Q(i) : y_p = y_i\}$ is the set of indices of all positives in the batch distinct from $i$, and $|P(i)|$ is its cardinality.

As shown in Figure 1, our final objective function is formulated as a weighted combination of a binary cross-entropy loss and contrastive loss.

$$L_{total} = \alpha L_{BCE} + L_{SCL} \quad (4)$$

## 3.1 Theoretical Analysis

Let $(X, Y, S) \sim p_{\text{data}}$, where $X \in \mathcal{X} \subset \mathbb{R}^{d_x}$ are features, $Y \in \{0, 1\}$ is the target, and $S \in \{0, 1\}$ is a binary sensitive attribute. An encoder $f_\theta : \mathcal{X} \to \mathcal{Z} \subset \mathbb{R}^{d_z}$ maps a sample to a representation $Z = f_\theta(X)$. Similarity between two representations is measured by $g_\tau(z, z') = \exp(\langle z, z' \rangle/\tau)$ with temperature $\tau > 0$.

**Positive-pair sampler.** Given an anchor sample $(x, y, s)$ with $y = 1$ (favourable label), we draw the positive according to the mixture

$$p_{\text{pos}} = \pi \, p_{\text{cross}} + (1 - \pi) \, p_{\text{within}}, \quad (5)$$

where

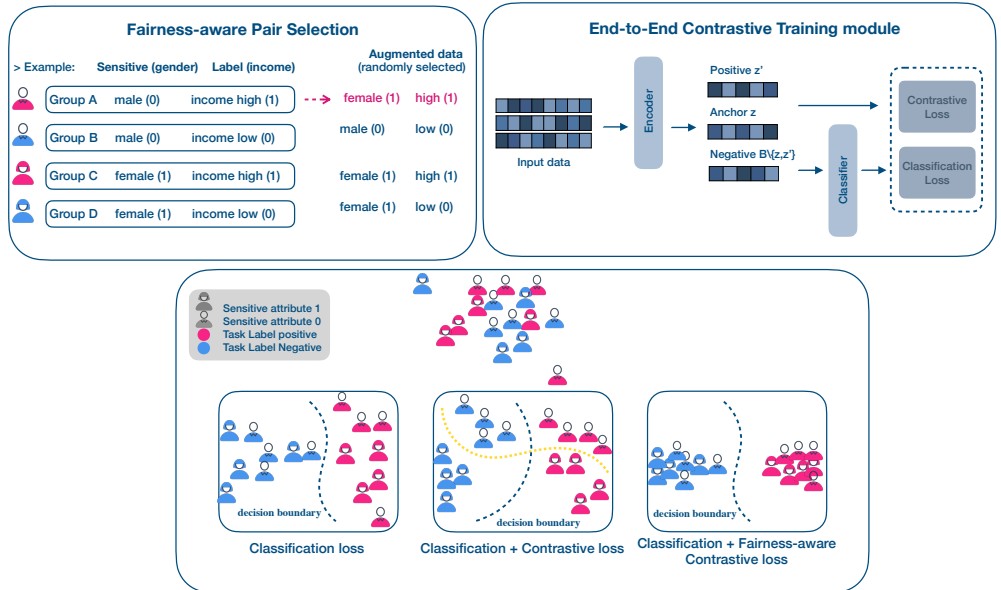

Figure 1: The schematic diagram illustrates the proposed fairness-aware contrastive learning framework. Our approach involves selectively sampling positive pairs based on specific criteria and integrating them into the training process with a contrastive loss in an end-to-end manner. Although combining supervised contrastive learning with cross-entropy loss improves model robustness, contrastive loss without explicit bias mitigation can unintentionally separate instances across sensitive attributes in the representation space. Our proposed fairness-aware contrastive loss, together with cross-entropy, reduces this separation by bringing positive-class instances from different sensitive groups closer, thereby improving fairness without requiring additional fairness-specific constraint loss functions.

- $p_{\text{cross}}\big((x, y, s), (x^+, y^+, s^+)\big)$ places probability mass only on pairs satisfying $y^+ = y = 1$ and $s^+ \neq s$;

- $p_{\text{within}}$ places mass on pairs with $y^+ = y = 1$ *and* $s^+ = s$.

Consequently, $\pi = \Pr[S^+ \neq S \mid Y = 1]$ is determined entirely by the data.

**Contrastive loss.** With one positive and $K$ negatives, the InfoNCE loss is

$$\mathcal{L}_{\text{NCE}}(\theta) = -\mathbb{E}_{(x, x^+) \sim p_{\text{pos}}}\left[\log \frac{g_\tau(z, z^+)}{g_\tau(z, z^+) + \sum_{k=1}^{K} g_\tau(z, z_k^-)}\right]. \tag{6}$$

Let $C \in \{\text{cross}, \text{within}\}$ be the indicator

$$C = \begin{cases} \text{cross} & \text{if } S^+ \neq S, \\ \text{within} & \text{if } S^+ = S \end{cases} \tag{7}$$
$$\Rightarrow \quad \Pr[C = \text{cross}] = \pi, \quad \Pr[C = \text{within}] = 1 - \pi.$$

**Lemma 3.1** (InfoNCE lower bound [32]). *For any encoder $f_\theta$ and any positive-pair distribution,*

$$-\mathcal{L}_{\text{NCE}}(\theta) + \log K = \underbrace{I(Z; Z^+)}_{\textit{MI of pairs}} - \underbrace{D_{\text{KL}}\big(p_{Z, Z^+} \,\|\, p_Z p_{Z^+}\big)}_{\geq 0} \tag{8}$$
$$\leq I(Z; Z^+).$$

*Thus minimising $\mathcal{L}_{\text{NCE}}$ maximises the mutual information $I(Z; Z^+)$.*

**Assumptions (for this proposition only).**

1. (*Pair-wise Markov property*)

$$Z \perp\!\!\!\perp Z^+ \;\middle|\; \begin{cases} Y & \text{if } C = \text{cross,} \\ (Y,S) & \text{if } C = \text{within.} \end{cases}$$

2. $C$ is a *deterministic* function of $(S, S^+)$; hence $C \perp\!\!\!\perp Z \mid (Y,S)$ and $C \perp\!\!\!\perp Z^+ \mid (Y,S)$.

**Proposition 3.2** (Mutual-information decomposition). *Under Assumptions 1–2,*

$$I(Z; Z^+) \;=\; I(Z; Y) \;+\; (1 - \pi)\, I(Z; S \mid Y). \tag{9}$$

*Proof (chain rule only).* Start from the law of total expectation for mutual information:

$$I(Z; Z^+) = \underbrace{I(Z; Z^+ \mid C)}_{\text{case analysis}} + \underbrace{I(Z; C)}_{\text{=0 by (Assumption 2)}} - \underbrace{I(Z; C \mid Z^+)}_{\text{=0 by (Assumption 2)}}. \tag{10}$$

Because $I(Z; C) = 0$ and $I(Z; C \mid Z^+) = 0$, only the first term remains. Expand it with the definition of conditional mutual information:

$$\begin{aligned} I(Z; Z^+ \mid C) = {} & \pi \cdot I(Z; Z^+ \mid C = \text{cross}) \\ & + (1 - \pi) \cdot I(Z; Z^+ \mid C = \text{within}). \end{aligned} \tag{11}$$

the cross-$S$ branch:
When $C = \text{cross}$, Assumption 1 gives the Markov chain $Z \perp\!\!\!\perp Z^+ \mid Y$. By the chain rule,

$$I(Z; Z^+ \mid C = \text{cross}) = I(Z; Y \mid C = \text{cross}) \quad (Z \perp\!\!\!\perp Z^+ \mid Y). \tag{a}$$

the within-$S$ branch:
When $C = \text{within}$, the Markov chain is $Z \perp\!\!\!\perp Z^+ \mid (Y, S)$. A second application of the chain rule yields

$$\begin{aligned} & I(Z; Z^+ \mid C = \text{within}) \\ & = I(Z; Y, S \mid C = \text{within}) \\ & = I(Z; Y \mid C = \text{within}) + I(Z; S \mid Y, C = \text{within}). \end{aligned} \tag{b}$$

dropping the $C$-condition inside the MI terms. Because $C$ is a function of $(S, S^+)$ and is independent of $Z$ once $(Y, S)$ is fixed (Assumption 2), conditioning on $C$ adds no information beyond $(Y, S)$:

$$\begin{aligned} I(Z; Y \mid C) &= I(Z; Y), \\ I(Z; S \mid Y, C = \text{within}) &= I(Z; S \mid Y). \end{aligned} \tag{c}$$

Plugging (a)–(c) into the weighted sum gives

$$\begin{aligned} I(Z; Z^+) &= \pi\, I(Z; Y) + (1 - \pi)\big[\, I(Z; Y) + I(Z; S \mid Y)\big] \\ &= I(Z; Y) \;+\; (1 - \pi)\, I(Z; S \mid Y), \end{aligned}$$

which is exactly (9). $\qquad\square$

**Theorem 3.3** (InfoNCE $\Longleftrightarrow$ information bottleneck). *Let $\lambda := 1 - \pi$. Under the assumptions of Proposition 3.2,*

$$\mathit{argmin}_\theta\, \mathcal{L}_{\text{NCE}}(\theta) \;=\; \mathit{argmax}_\theta \Big\{ I(Z; Y) - \lambda\, I(Z; S \mid Y) \Big\}. \tag{12}$$

*Proof.* Combine Lemma 3.1 and the exact identity (9):

$$\begin{aligned} \mathcal{L}_{\text{NCE}}(\theta) &= -I(Z; Z^+) + \log K \\ &= -I(Z; Y) - (1 - \pi)I(Z; S \mid Y) + \log K. \end{aligned} \tag{13}$$

Since $\log K$ is constant in $\theta$, minimising $x\mathcal{L}_{\text{NCE}}$ is equivalent to maximising the right-hand side of (12). $\qquad\square$

Table 1: Hyperparameter configuration used for training on the Adult, Health, and German datasets. All models were optimized using the Adam optimizer with a fixed learning rate and temperature for the contrastive loss.

| Hyperparameter | Adult | Health | German |
|---|---|---|---|
| Encoder hidden layers | [64, 64, 64] | [128, 64, 64] | [32, 32, 32] |
| Classifier hidden layers | [16] | [16] | [16] |
| Epochs | 100 | 100 | 100 |
| Contrastive loss temperature ($\tau$) | 1 | 1 | 1 |
| Learning rate | $1 \times 10^{-3}$ | $1 \times 10^{-3}$ | $1 \times 10^{-3}$ |
| Optimizer | Adam | Adam | Adam |

**Corollary 3.3.1.** *The hyper-parameter $\lambda$ that trades off predictive utility $I(Z;Y)$ against conditional leakage $I(Z;S \mid Y)$ is* entirely data-driven: *large values of $\pi$ (many cross-group pairs) automatically reduce the penalty on $I(Z;S \mid Y)$ and vice-versa.*

**Implications.** Equation (12) shows that our *pair-selection policy alone* turns standard contrastive learning into an **information bottleneck** that *(i)* preserves label-relevant bits and *(ii)* suppresses sensitive bits *conditioned* on the label. Unlike adversarial critics or explicit MI estimators, no additional modules or tunable coefficients are needed; fairness–accuracy trade-offs emerge directly from the data distribution. Empirically, we observe that increasing $\pi$ tightens demographic-parity and equal-opportunity gaps while maintaining task performance, corroborating the theoretical guarantee.

# 4 Experiments

We provide the full experimental setup details, including model architecture and training hyperparameters, in Table 1. These configurations were selected through empirical tuning based on validation performance. The model training was performed using an NVIDIA GeForce RTX 3090 GPU.

## 4.1 Datasets.

We validate our model on three benchmark datasets in the fairness domain.

- **Adult.** [12] This dataset contains demographic data of 48,842 individuals, and the main task is to predict whether the income of the individual is greater than 50k or not. The sensitive attribute is gender.

- **German Credit** [12] This dataset consists of 1000 individuals and their banking information. The primary task is to predict an individual's credit in repaying their loan. The sensitive attribute is age. Consistent with the setup in [13], we binarize the age feature into "younger" and "older" groups, treating the "older" group as the privileged class. The threshold of 25 years is chosen based on findings from [25], which identified this split as having the highest potential for discriminatory impact.

- **Heritage Health**[1] This dataset contains information of about 50k patients and their corresponding medical conditions. The task is to predict the Charlson Comorbidity Index, a 10-year mortality risk index. The sensitive attribute is age, which has been categorized into nine values. Prior analyses have shown that the dataset exhibits bias against older individuals.

## 4.2 Evaluation Metrics.

Various fairness notions have been defined and utilized in the fairness domain. Group fairness, including metrics such as Demographic Parity (DP), Equalized Odds (EO), and Equal Opportunity (EOPP) [20, 41], is a commonly used concept in the literature. In this study, we adopt **Demographic Parity (DP)**, also known as statistical parity, as our main fairness metric. Demographic parity ensures that the probability of receiving a favorable outcome is being equitably distributed across groups(privileged and unprivileged). Specifically, this metric requires that the likelihood of all

---

[1]https://foreverdata.org/1015/index.html

positive predictions (both true positives and false positives) be similar across these groups. Thus, discrimination or disparities can be quantified by measuring the difference between the conditional probabilities of positive predictions for the privileged and unprivileged groups:

$$P(\hat{Y} = 1 \mid X, S = 1) = P(\hat{Y} = 1 \mid X, S = 0) \tag{14}$$

### 4.3 Baselines.

We evaluate our proposed model, FairContrast, in both unsupervised and self-supervised settings and compare our method with various baselines:

- **Base MLP** We train an MLP classifier without incorporating any fairness measures as our biased base model.

- **FCRL** [16] A fair representation learning framework that introduces a robust method to control parity using mutual information based on contrastive information estimators. By constraining mutual information between representations and sensitive attributes, [16] controls the parity of any downstream classifier.

- **CVIB** [31] A fair representation learning framework that proposes a conditional variational autoencoder to derive representations invariant to sensitive attribute. Their approach is based on a single information-theoretic optimization without adversarial training.

- **Adversarial forgetting** [24] A novel representation learning framework for invariance induction through the "forgetting" mechanism as an information bottleneck to learn invariant representations.

- **Counterfactual Data Augmentation** To demonstrate the importance of data augmentation in contrastive learning, we also conducted a comparison between our method and counterfactual data augmentation. This approach is based on counterfactual fairness [28], where a decision is considered fair if it is the same in both the actual world and the counterfactual world. We generate counterfactual data points by converting the sensitive attribute to counterfactual values, while leaving all other attributes unchanged. We then integrate this data augmentation into our supervised contrastive learning framework. This data aumentation is only applicable on binary sensitive attributes.

- **SCARF** [3] As discussed earlier, SCARF is a state-of-the-art contrastive learning approach on tabular data. They mask 60% of the features and use Random Feature Corruption to replace these masked values. They deploy these data augmentations as positive pairs in their contrastive learning framework. They use InfoNCE as their loss function.

- **VIME** [45] As discussed earlier, VIME is a state-of-the-art self-supervised framework for tabular data. They pre-train an encoder on unlabeled masked data to extract representations. They used Bernoulli distribution to randomly generate the mask and the CutMix method [46] to fill the masked value. We evaluated this model in both semi- and self-supervised settings.

### 4.4 Experimental Results.

The trade-off between accuracy and fairness across three datasets is shown in Figure 2. The optimal region of the graph is in the lower right corner, indicating higher accuracy and fair outcome (lower demographic parity). Similarly to [16], the results reported for various benchmarks are average accuracy and maximum demographic parity over five runs with random seeds. As evident from the figures, in the Adult dataset, our FairContrast-supervised model stands out, achieving the highest accuracy within the demographic parity range of $0 \sim 0.075$, demonstrating its ability to provide fair predictions while maintaining strong performance. Within the demographic parity range of $0.075 \sim 0.125$, our FairContrast-unsupervised model outperforms others, further showcasing the robustness of our framework even without supervision. In contrast, models such as VIME and SCARF, which are not explicitly designed to address fairness, exhibit a higher bias in their results, reflected by higher DP values, similar to the unfair MLP. For the German dataset, both the FairContrast-supervised and FairContrast-unsupervised models continue to demonstrate superior performance, particularly in the demographic parity range of $0 \sim 0.05$. Comparatively, models such as Adversarial Forgetting and

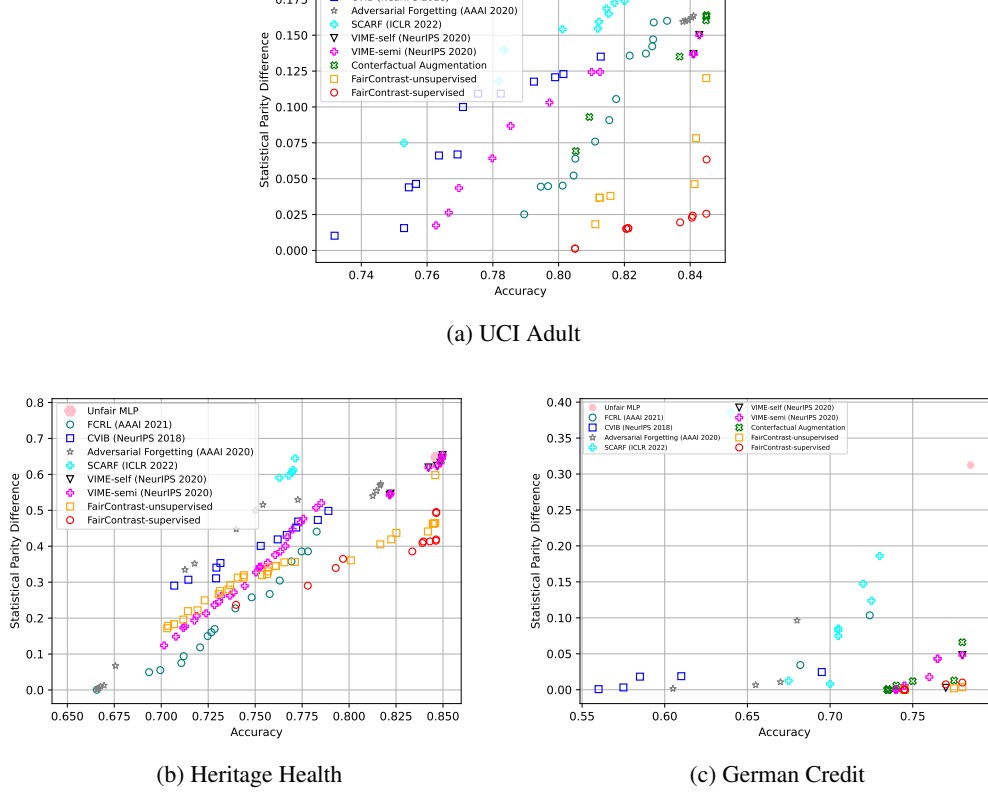

(a) UCI Adult

(b) Heritage Health

(c) German Credit

Figure 2: Accuracy-fairness trade-off and comparison to various benchmark models across three benchmark datasets: (a) UCI Adult dataset, (b) Heritage Health dataset, and (c) German Credit. The optimal region on the graph is the lower right corner, representing high accuracy and low demographic parity. Our model demonstrates a superior fairness-accuracy trade-off.

CVIB achieve lower DP values, but at the cost of significant accuracy loss. VIME model shows less bias in this dataset, as indicated by their position on the graph. In the Health dataset, our FairContrast-supervised model achieves the highest accuracy within the demographic parity range of $0.3 \sim 0.5$, confirming its effectiveness in providing fair and accurate predictions even in the more challenging dataset. Our FairContrast-unsupervised model shows comparable performance within this range, further underscoring the versatility of our approach. When focusing on the demographic parity range of $0.2 \sim 0.3$, both our supervised model and FCRL exhibit comparable accuracy, indicating that FCRL is also competitive in this particular fairness range. However, models like SCARF and VIME again demonstrate higher bias, as reflected by their positions further up in the DP range. Across all three datasets, our FairContrast models, both supervised and unsupervised, consistently occupy the optimal region of the trade-off graphs, balancing high accuracy with low demographic parity difference. This confirms the effectiveness of our approach in achieving fairness without compromising performance. In contrast, state-of-the-art models like VIME and SCARF, which do not explicitly target fairness, exhibit bias levels similar to the Unfair MLP, as evidenced by their higher DP values across the datasets. This analysis highlights the robustness and effectiveness of our FairContrast framework to ensure that models not only perform well, but also adhere to fairness constraints, making it a valuable contribution to the field of fair representation learning.

For quantitative comparison, we also report the best accuracy corresponding to the worst-case scenario of demographic parity results for all models on the three datasets, summarized in Table 2. Our proposed FairContrast-supervised model consistently demonstrates superior performance across all three datasets—Adult, German, and Health—achieving the best or nearly the best results in both accuracy and fairness (lowest DP). Specifically, in the Adult dataset, FairContrast-supervised achieved

Table 2: Accuracy and demographic parity (DP) on three benchmark datasets. Lower DP indicates higher fairness. Gray highlights the baseline model (Unfair MLP), Green highlights our method (FairContrast), **bold** marks the best performance among all methods, and *italic* denotes the second-best.

| Dataset | Model | Accuracy ↑ | DP ↓ |
|---|---|---|---|
| **Adult** | Unfair MLP | 84.5 | 0.1855 |
| | FCRL | 83.29 | 0.16 |
| | CVIB | 81.28 | 0.1350 |
| | Adversarial forgetting | 84.1 | 0.1635 |
| | Counterfactual | 84.49 | 0.1639 |
| | VIME-semi | 84.27 | 0.15 |
| | VIME-self | 84.47 | 0.1779 |
| | SCARF | 82.13 | 0.1848 |
| | **FairContrast (Ours)-unsupervised** | **84.4** | *0.1201* |
| | **FairContrast (Ours)-supervised** | **84.4** | **0.0255** |
| **German** | Unfair MLP | 78.5 | 0.3125 |
| | FCRL | 72.4 | 0.1035 |
| | CVIB | 69.5 | *0.0244* |
| | Adversarial forgetting | 68 | 0.0963 |
| | Counterfactual | 78 | 0.066 |
| | VIME-semi | 76.5 | 0.0431 |
| | VIME-self | 78 | 0.0482 |
| | SCARF | 73 | 0.1862 |
| | **FairContrast (Ours)-unsupervised** | **78** | 0.0297 |
| | **FairContrast (Ours)-supervised** | **78** | **0.0099** |
| **Health** | Unfair MLP | 84.64 | 0.6468 |
| | FCRL | 78.27 | *0.4407* |
| | CVIB | 78.9 | 0.4982 |
| | Adversarial forgetting | 81.68 | 0.5733 |
| | VIME-semi | 82.2 | 0.5463 |
| | VIME-self | 84.22 | 0.6192 |
| | SCARF | 77.12 | 0.6444 |
| | **FairContrast (Ours)-unsupervised** | *84.19* | 0.4410 |
| | **FairContrast (Ours)-supervised** | **84.3** | **0.4135** |

an accuracy of 84.4 % with a DP of 0.0255, indicating a substantial reduction in bias compared to other models. Similarly, in the German dataset, our model maintained strong accuracy at 78 % while achieving the lowest DP of 0.0099, further confirming its ability to mitigate bias effectively. Our proposed unsupervised model also performs well, with relatively low DP scores compared to other models, though its accuracy is slightly lower than that of the FairContrast-supervised model. Although other models like FCRL and CVIB offer competitive alternatives, particularly in fairness, they often fall short in achieving the same level of accuracy or in minimizing bias as effectively as FairContrast. State-of-the-art models, such as VIME and SCARF, which are not specifically focused on enhancing fairness, achieve accuracy comparable to our supervised model. However, the bias in their representations is similar to that found in the unfair MLP model. Overall, our FairContrast framework represents a significant advancement in contrastive learning for tabular data, offering a robust solution that does not compromise on fairness while maintaining strong predictive performance.Our results suggest that contrastive learning, when properly supervised and designed with fairness in mind, can lead to models that perform well both in terms of accuracy and fairness.

## 4.5 Ablation on Classification Loss Weight

To further analyze the impact of the loss weight $\alpha$ on the fairness–accuracy trade-off, we conduct an ablation study using the Adult dataset. Specifically, we evaluate the Area Over the Fairness–Accuracy Pareto Curve (AOC) at varying values of $\alpha$ in both supervised and unsupervised settings. The AOC summarizes the feasible region in the parity–accuracy space and offers a quantitative proxy for a method's capacity to provide accurate predictions under fairness constraints. A higher AOC indicates that a method can achieve better utility while satisfying a wider range of demographic parity thresholds.

Following the interpretation presented in Gupta et al. [16], the parity–accuracy curve reflects the achievable frontier between accuracy and fairness, where methods that shift the curve closer to the bottom right are more desirable. Thus, the area under this frontier—the AOC—represents the

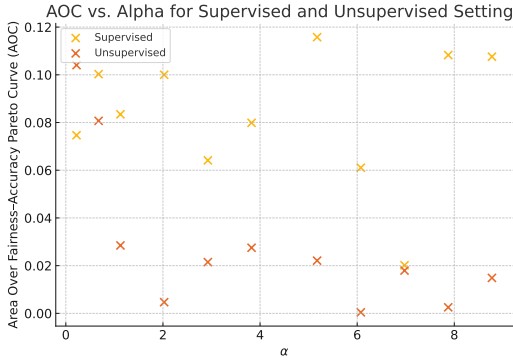

Figure 3: Effect of varying $\alpha$ on the Area Over the Fairness–Accuracy Pareto Curve (AOC) for supervised and unsupervised settings. Each point represents the AOC score at a specific $\alpha$ value. The trade-off stabilizes for $\alpha > 1$, indicating consistent fairness–accuracy performance in both learning modes.

volume of favorable outcomes. In our results (Fig. 3), performance stabilizes across both learning settings when $\alpha > 1$, suggesting that moderate weighting of the classification loss produces robust representations with respect to both utility and fairness.

## 5 Conclusion

Contrastive learning has shown its effectiveness in improving model robustness and generalizability across various domains, including Natural Language Processing (NLP), computer vision, and speech recognition. Recently, there has been an increasing interest in applying self-supervised contrastive learning to tabular data. Although handling data types such as images, text, or speech is less challenging due to their feature correlations, semantic relationships, and structured information, tabular datasets pose unique challenges due to the lack of explicit relationships within their features, which can vary across different datasets.

In this paper, we argue that current state-of-the-art models for tabular data, such as VIME and SCARF, do not address fairness issues. The fairness of the learned representations has not been thoroughly examined, and these models exhibit biases in their predictions, leading to discrimination in the downstream tasks. To address this, we propose supervised and self-supervised contrastive learning frameworks for tabular data to mitigate bias and improve fairness. Our approach involves selective pairing of samples based on specific criteria and incorporating these pairs into the training process with a contrastive loss. This method encourages the embeddings of paired instances to be closer together, reducing discrimination based on sensitive attributes.

We evaluated our proposed method using three benchmark datasets in the fairness domain. The results show a significant reduction in bias compared to existing state-of-the-art frameworks for tabular data. Furthermore, these fair representations can be applied to any downstream tasks.

Although our framework achieves promising results, several limitations should be noted.

First, the work mainly addresses group fairness through metrics such as Demographic Parity. While these are useful for capturing disparities between subgroups, they do not fully account for individual fairness, which ensures that similar individuals are treated similarly. Future research could explore approaches that jointly address both group and individual level fairness.

Second, our method currently emphasizes demographic parity. In real-world scenarios, multiple fairness definitions may be relevant, and these can sometimes conflict with one another. Extending the framework to accommodate several fairness criteria simultaneously would increase its practical flexibility.

Third, although our approach was designed with tabular data in mind, the underlying methodology could also be extended to other data types, including images, text, or multimodal systems. Exploring these extensions remains an open avenue for future work.

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
