# OpenReview forum: "FairContrast: Enhancing Fairness through Contrastive learning and Customized Augmenting Methods on Tabular Data"
_NeurIPS.cc/2025/Workshop/Reliable_ML — NeurIPS 2025 - Reliable ML Workshop_

### Official Review · Reviewer_TJZd · 2025-09-15
**Decent contribution, page limit exceeded**

**Rating:** 4
**Confidence:** 3

**Review:**

**Note** This paper seemed to exceed the 9-page limit stipulated on the workshop website, which is unfortunate (my rating considers this fact) as I think this paper could have been an interesting contribution to achieving fairness via contrastive learning in tabular data. This could have been fixed by placing proofs into the appendix and only keeping proof ideas in the main sections.

*Summary*

This paper studies how to incorporate contrastive learning to achieve fairness. It first identifies the literature doing fair contrastive learning, and claim that this is underexplored for tabular data. The methodology (FairContrast) is that they train via a combination of both the binary cross-entropy loss and contrastive loss, partitioned into self-supervised and supervised contrastive loss (former: any pair of data are negative examples unless they are random augmentation of each other; latter: positive samples are those belonging to the same class). The "intuition" of how fairness is achieved is where for the same label, one augments data by randomly flipping the sensitive attributes (e.g. gender). The paper also provides theoretical analysis by lower-bouding the InfoNCE loss via information bottleneck (Lemma 3.1, Theorem 3.3). In the experimental section, the paper considers a few baselines, without (MLP) and with contrastive learning (FCRL, CVIB, SCARF, etc), and demonstrated that their FairContrast paper can outperform most other works by showing a low demographic parity  without sacrificing much accuracy score. In the ablation studies, this work also demonstrates that the AOC-tradeoff stabalizes for sufficiently large $\alpha$ value on the BCE loss.

*Strengths*

This work is a good addition to incorporating fairness to contrastive learning. The loss function is well-thought (with good intuition of why BCE and contrastive loss), and also backed by theoretical analysis of why this loss is equivalent to information bottleneck. In addition, the experimental section shows a thorough comparison with a few baselines, including some SOTA ones like SCARF.

*Weaknesses*

One main question I have is the claim of why contrastive learning is harder on tabular data. Precisely:

- what are some limitations if one uses some existing frameworks for tabular data (e.g. TabPFN, "Accurate predictions on small data with a tabular foundation model")?

- In lines 31-33 the authors explain the lack of structured information in tabular data. However, isn't this relationship difficult to learn even for images/text/speech data (despite those information being present)?

Several presentation suggestions that can improve the paper:

- Line 247: do you mean $P(Y=1|X,S=1)-P(Y=1|X,S=0)$? (Also shouldn't be absolute value of the difference?)

- Lines 195, 197: "The cross-S branch" wording should be made clear that you're referring to a subsection (subcase) within the proof.

---

### Official Review · Reviewer_eJBz · 2025-09-19
**Review of submission #118**

**Rating:** 7
**Confidence:** 2

**Review:**

This paper introduces FairContrast, a fairness-aware contrastive learning framework for tabular data. The core idea is to modify the positive sampling strategy in contrastive learning: privileged samples with favorable outcomes are explicitly paired with unprivileged samples of the same outcome. This encourages representations that are predictive but less dependent on sensitive attributes. The method is theoretically motivated via an information bottleneck interpretation, and empirically validated on three fairness benchmark datasets.

* Strengths:

1) The fairness-aware sampling strategy is conceptually straightforward and easy to implement, yet yields significant improvements.
2) Across the three datasets, it seems that the method consistently improves fairness while maintaining accuracy, outperforming both fairness-aware and contrastive baselines.
3) Another key strength is that the sampling probability $\pi$ has a very simple form and it's data-driven. It's not a tunable hyperparameter but instead emerges from the data.

* Weaknesses
1) The writing feels a bit dense in some parts. In particular, I would appreciate a high-level roadmap in Section 3.1 and a highlight of the experimental results in the main body with most details in an Appendix.
2) It would be good if you could report error bars (and standard deviations in the tables) across independent executions of the experiments

Notes:
Please fix the identation of the equation in lines 199-200.